## TRANSLATIONAL PERSPECTIVE

# A shock to the system: neurostimulation therapy for opioid-induced respiratory depression

Ken D. O'Halloran 

*Department of Physiology, School of Medicine, College of Medicine & Health, University College Cork, Cork, Ireland*

Email: k.ohalloran@ucc.ie

Edited by: Scott Powers

Linked articles: This Translational Perspective article highlights an article by Huang et al. To read this paper, visit https://doi.org/10.1113/JP282664.

The peer review history is available in the Supporting information section of this article (https://doi.org/10.1113/JP283272#support-information-section).

Neurostimulation strategies are currently employed therapeutically in several wide-ranging conditions such as intractable pain, urinary and faecal incontinence, and several refractory neurological and psychiatric disorders. Neurostimulation therapy is centred on the use of invasive or non-invasive electrical stimulation to drive neural activation, thereby modifying neural circuitry behaviour. From a relatively crude base many decades ago, contemporary approaches have evolved to the use of customizable control devices for optimal clinical outcomes. Notwithstanding common usage, and proven clinical efficacy, gaps remain in many instances, in the fundamental understanding of the mechanisms underpinning effective neuromodulation therapies. Nevertheless, the impressive gains thus far and the perpetual ambition to break new ground have fostered considerable interest in widening the portfolio of potential application.

Spinal cord stimulation is employed in the treatment of chronic refractory pain and has shown some promise in the restoration of function in people with spinal cord injuries. Respiratory motor nuclei are housed in the spinal cord with efferent motor projections to various muscles of breathing. Some disorders marked by ventilatory insufficiency can be addressed by diaphragm muscle pacing (Bach, 2013), although this may be contraindicated in some settings (Gonzalez-Bermejo et al., 2016). Whereas pacing is achieved using direct electrical stimulation along the efferent pathway outside of the spinal cord (primarily at the level of the target muscle), there are some scenarios where optimal intervention and/or logistical necessity may require activation of spinal circuits to promote respiratory behaviour. One striking scenario where this applies is naloxone-resistant opioid-induced respiratory arrest.

Synthetic opioid overdose can completely suppress respiratory rhythm and pattern formation, or in the event of manifestation of a neural substrate for breathing by brainstem networks, opioids can suppress the motor transduction of pre-motor neural activity by profoundly hyperpolarizing respiratory motor neurons (Ramirez et al., 2021). As such, depolarizing current to activate motor neurons can be an effective strategy to override the inhibitory effects of opioids. Respiratory rescue from opioid overdose has recently been demonstrated in a rodent model using minimally invasive temporal interference stimulation of the spinal cord, paving the way for human trials (Sunshine et al., 2021). In this issue of *The Journal of Physiology*, Huang et al. (2022) illustrate the efficacy of spinal cord stimulation in restoring respiratory effort during opioid-induced respiratory depression in humans.

In people undergoing surgery for degenerative spinal disease, breathing was suppressed or completely inhibited by the potent short-acting synthetic opioid analgesic, remifentanil, against the backdrop of propofol anaesthesia. Epidural stimulation of the posterolateral cervical spinal surface was shown, for some locations that were carefully mapped, to ameliorate opioid-induced respiratory depression and respiratory arrest. Interestingly, the stimulus was sub-threshold for motor activation and therefore is reasoned to have activated respiratory motor outflow circuitously via the activation of ascending spinobulbar pathways, evidenced by concurrent activation of cranial motor outflow to the tongue muscle. Whilst this reasonably explains the capacity to enhance opioid induced *suppression* of respiratory motor outflow, presumably by increasing the excitability of motor pools to the prevailing remnant level of central respiratory drive, it does not readily explain the capacity for spinal neuromodulation to restore rhythmic respiratory airflow in the case of opioid-induced *elimination* of breathing (apnoea), when descending respiratory outflow is predominantly quiescent (expiratory motor activation is revealed in some respiratory-related pathways during central apnoea). Thus, intriguingly, the capacity for spinal stimulation to drive cyclical respiratory airflow during opioid-induced central apnoea may point to unexplored circuitry and perhaps the capacity for spinal circuitry to drive respiratory rhythm, albeit with significantly curtailed ventilatory capacity (low tidal volume) at least in response to the stimulus paradigm employed in the study (Huang et al., 2022).

These proof-of-principle observations in a controlled surgical setting are very encouraging for the notion of minimally invasive spinal stimulation for opioid-induced respiratory depression in the context of acute paramedicine. Opioid overdose is commonly fatal. Point-of-care use of a tailored stimulation device could potentially rescue breathing, at least sufficiently for secondary support such as oxygen supplementation to prevent profound hypoxia. Enabling technologies might further refine neuromodulation strategies to provide full rescue to breathing providing a life-saving device, analogous to the cardiac defibrillator. Depending upon the mechanism of action, there is also the potential for such devices to support ventilation in other settings such as spinal cord injury and neurodegenerative conditions characterized by ventilatory insufficiency. It is interesting to consider in such settings, the potential to develop interventional paradigms that might drive plasticity in respiratory neural circuits, either alone or in combination with other task-specific rehabilitative strategies.

Reverse translation of the findings by Huang et al. (2022) to pre-clinical experimental models will be an important step in the delineation of the fundamental circuitry and mechanisms at the root of the exciting and important findings. In that sense, the study is a timely reminder of the central role of physiology in rehabilitation-focused clinical specialties that are centred on patient care and improved quality of life.

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

## Additional information

### Competing interests

None declared.

### Author contributions

Sole author.

### Funding

None.

### Acknowledgements

### Keywords

apnoea, neurostimulation, remifentanil, spinal cord

## Supporting information

Additional supporting information can be found online in the Supporting Information section at the end of the HTML view of the article. Supporting information files available:

**Peer Review History**

