## [Peer Review History · The Journal of Physiology]

A shock to the system: neurostimulation therapy for opioid induced respiratory depression

Ken D O'Halloran

DOI: 10.1113/JP283272

Corresponding author(s): Ken O'Halloran (k.ohalloran@ucc.ie)

The following individual(s) involved in review of this submission have agreed to reveal their identity: Daniel C Lu (Referee #1)

Review Timeline:

Submission Date:

27-Apr-2022

Accepted:

04-May-2022

Senior Editor: Scott Powers

Reviewing Editor: Scott Powers

Transaction Report:

Dear Professor O'Halloran,

Re: JP-TP-2022-283272 "A shock to the system: neurostimulation therapy for opioid induced respiratory depression" by Ken D O'Halloran

I am pleased to tell you that your invited Translational Perspective has been accepted for publication in The Journal of Physiology, subject to any modifications that may be required by the Journal Office to conform to House rules.

NEW POLICY: In order to improve the transparency of its peer review process The Journal of Physiology publishes online as supporting information the peer review history of all articles accepted for publication. Readers will have access to decision letters, including all Editors' comments and referee reports, for each version of the manuscript and any author responses to peer review comments. Referees can decide whether or not they wish to be named on the peer review history document.

The last Word version of the paper submitted will be used by the Production Editors to prepare your proof. When this is ready you will receive an email containing a link to Wiley's Online Proofing System. The proof should be checked and corrected as quickly as possible.

All queries at proof stage should be sent to tjp@wiley.com

Thank you for your contribution to The Journal of Physiology.

Yours sincerely,

Scott K. Powers
Senior Editor
The Journal of Physiology
<https://jp.msubmit.net>
<http://jp.physoc.org>
The Physiological Society
Hodgkin Huxley House
30 Farringdon Lane
London, EC1R 3AW
UK
<http://www.physoc.org>
<http://journals.physoc.org>

Editor Comments:

Thank you for this outstanding contribution to the Journal of Physiology.

Reviewer 1 Comments:

We appreciate both perspectives that you solicited. They enrich the work and provide useful counterpoints, which we will explore in future studies.